# Enhanced Photo-Assisted Fenton Degradation of Antibiotics over Iron-Doped Bi-Rich Bismuth Oxybromide Photocatalyst

**DOI:** 10.3390/nano13010188

**Published:** 2022-12-31

**Authors:** Fengjiao Zhang, Yanhua Peng, Xiaolong Yang, Zhuo Li, Yan Zhang

**Affiliations:** School of Chemistry and Chemical Engineering, Qingdao University, Qingdao 266071, China

**Keywords:** Bi_4_O_5_Br_2_ nanosheet, element doping, photo-Fenton, tetracycline, carrier separation kinetics

## Abstract

Herein, combining photocatalysis and Fenton oxidation, a photo-assisted Fenton system was conducted using Fe-doped Bi_4_O_5_Br_2_ as a highly efficient photocatalyst to realize the complete degradation of Tetracycline antibiotics under visible light. It has been observed that the optimized photocatalyst 5%Fe-doped Bi_4_O_5_Br_2_ exhibits a degradation efficiency of 100% for Tetracycline with H_2_O_2_ after 3 h visible-light irradiation, while a degradation percentage of 59.8% over the same photocatalyst and 46.6% over pure Bi_4_O_5_Br_2_ were obtained without the addition of H_2_O_2_ (non-Fenton process). It is unambiguous that a boost photo-assisted Fenton system for the degradation of Tetracycline has been established. Based on structural analysis, it demonstrated that the Fe atoms in place of the Bi sites may result in the distortion of the local structure, which induced the occurrence of the spontaneous polarization and thus enhanced the built-in electric field. The charge separation efficiency is enhanced, and the recombination of electrons and holes is inhabited so that more charges are generated to reach the surface of the photocatalyst and therefore improve the photocatalytic degradation efficiency. Moreover, more Fe (II) sites formed on the 5%Fe-Bi_4_O_5_Br_2_ photocatalyst and facilitated the activation of H_2_O_2_ to form oxidative species, which greatly enhanced the degradation efficiency of Tetracycline.

## 1. Introduction

Antibiotics are quite effective in resisting microorganisms and treating biological diseases [1,2]. Tetracycline (TC), the quintessential antibiotic, is commonly employed, possessing the broadest antibacterial spectrum as well as a tough chemical structure [3]. However, the massive misuse of TC must be accompanied by incomplete metabolism and accidental spills due to its intrinsic toxicity and constancy. Tiny TC can seriously poison the aquatic environment and conversely threaten life on Earth [4,5]. Therefore, removing tetracycline and other antibiotics has become one of the important research topics in the environmental field.

The photocatalytic technique as an advanced oxidation processes (AOPs) stands out among various technologies to purify antibiotic sewage, which is apparently a “green” process driven by sustainable solar energy [6]. It is presently significant to harvest visible light (at least 50% of the solar spectrum) for efficient photocatalytic elimination of TC. As a novel photocatalyst, bismuth oxybromide (BiOBr) exhibits potential applications on energy generation, environmental remediation, bacterial disinfection and so on, particularly in wastewater treatment [7,8]. Pristine BiOBr possesses a bandgap of ~2.8 eV determined by a sandwich-like molecular structure with [Bi_2_O_2_]^2+^ and a double layer of Br; however, the weak visible-light harvest and slow carrier separation efficiency hindered its wide applications [9]. To alleviate the above symptom, a “Bi-rich” strategy is employed to optimize the unique molecular structure, and a series of Bi-rich bismuth oxybromide (Bi_4_O_5_Br_2_, Bi_3_O_4_Br, Bi_24_O_31_Br_10_, Bi_5_O_7_Br, et al.) are employed [10,11,12,13,14]. The suitable bandgap (~2.5 eV) and the great charge transfer property of Bi_4_O_5_Br_2_ demonstrate more photogenerated carriers excited by visible light, which is used to modulate the valence band (VB, contributed by O 2p and Br 4p) and conduction band (CB, contributed by Bi 6p) potentials through enhancing the local structure [15]. Li et al. claimed that more superoxide radicals (^•^O_2_^−^) may be generated in the negative CB potential of Bi_4_O_5_Br_2_ compared to that of BiOBr, which is beneficial for the degradation of ciprofloxacin [16]. Thus, tuning the band structure and optimizing the carrier dynamics are of great importance to expedite a photocatalytic reaction.

It is worthy to mention that the band structure engineering of Bi_4_O_5_Br_2_ is also carried out via deliberately doping the foreign heteroatoms (anions, cations), which influences the potential difference due to the spontaneous polarization and internal electric field [17,18,19]. Therefore, the photogenerated carriers are effectively separated and transferred driven by the built-in electric field [19]. Wang et al. found that iodine, also in the halogen family, was doped in Bi_4_O_5_Br_2_, resulting in a negative shift of VB potential, and thus improved the separation of the carriers [20]. Thus, more ^•^O_2_^−^ and h^+^ generated from I_0.7_-Bi_4_O_5_Br_2_ attacked the parabens with a higher efficiency. Zhang et al. studied the photocatalytic NO removement of Mn-doped Bi_4_O_5_Br_2_ and observed a superior photocatalytic oxidation activity owing to the excellent O_2_ capture ability and the strong oxidative hydroxyl radical (^•^OH) generated from the photoelectrons [21]. However, unfortunately it is still a critical task for an entire degradation of TC contained in wastewater.

Recently, a “green” process that combines the advanced photocatalytic technique with the intensive Fenton oxidation process (heterogeneous photo-assisted Fenton oxidation) was reported [22,23,24], by which an entire degradation of TC can be achieved efficiently [25,26]. In this photo-Fenton process, more photogenerated charges shifted onto the photocatalyst surface and played a decisive role on the degradation efficiency of organic pollutants because more charges may activate the iron ions by reacting with H_2_O_2_ to produce more ^•^O_2_^−^ and ^•^OH [27,28,29]. The GO-FePO_4_ was first employed as a heterogeneous photo−Fenton catalyst to the degradation of rhodamine B by Yu et al., which reported the H_2_O_2_ could be easily activated under solar energy excitation [30]. Peng et al. introduced Fe into g-C_3_N_4_ and created a Fe-g-C_3_N_4_/Bi_2_WO_6_ photocatalyst, which is able to activate H_2_O_2_ to facilitate the degradation of TC [31]. Zhou et al. claimed the doping of Fe into Rectorite not only showed the quick decomposition performance of H_2_O_2_ but also enhanced the charge transfer on the material [32]. Cheng et al. prepared a LaFeO_3_/BiOI heterojunction photocatalyst, combining the superiority of charge separation in the heterojunction and advanced Fenton activity of LaFeO_3_ [33]. Thus, the photogenerated electrons are transferred to Fe (II)/Fe (III) and activated H_2_O_2_, which makes the degradation of organic pollutants [34,35,36].

In this manuscript, Fe-doped Bi_4_O_5_Br_2_ nanosheets were synthesized to enhance the degradation of TC under visible-light irradiation. The successful doping of Fe ions resulted in the variation of the local structure distortion, which enhanced the spontaneous polarization and thus improved the built-in electric field. This improvement boosted the photons harvest and photocarriers separation. Moreover, the activation of H_2_O_2_ by the doped Fe^2+^/Fe^3+^ formed ^•^O_2_^−^ and ^•^OH. The dependence of photo-assisted Fenton oxidative antibiotics on the structure of the photocatalysts is investigated in detail.

## 2. Materials and Methods

### 2.1. Materials

All the chemical reagents used in the experiments, including bismuth nitrate pentahydrate, iron nitrate nonahydrate, potassium bromide, ethylene glycol, sodium hydroxide and ethanol absolute are analytically pure and purchased from Sinopharm Chemical Reagent Co., Ltd. (Shanghai, China) without any further purification.

### 2.2. Preparation of the Photocatalysts

The one-step solvothermal method was used for the preparation of all the photocatalysts in this study. The steps are illustrated as Figure 1: first, bismuth nitrate pentahydrate and iron nitrate nonahydrate were dissolved in ethylene glycol (25 mL) at various mass ratios (Bi:Fe = 99:1, 97:3, 95:5 and 93:7) and labeled as solution A, and 2 mmol potassium bromide was dissolved into 10 mL ethylene glycol, labeled as solution B. Then solution B was slowly added into solution A by stirring vigorously for 30 min until they were homogeneously mixed. The pH of the mixed solution was regulated to 10.5 using a 2.0 mol/L sodium hydroxide solution. Finally, the mixture was charged into a 100 mL Teflon-lined autoclave and kept at 160 °C for 12 h. The product was washed with deionized water and anhydrous ethanol 3 times, respectively, and then dried at 70 °C. The resulting products were labeled as Bi_4_O_5_Br_2_, 1%Fe-Bi_4_O_5_Br_2_, 3%Fe-Bi_4_O_5_Br_2_, 5%Fe-Bi_4_O_5_Br_2_ and 7%Fe-Bi_4_O_5_Br_2_, respectively.

### 2.3. Characterizations

Raman vibrational spectra were achieved on a DXR2 spectrometer (Thermo, Waltham MA, USA), which was excited by a 532 nm laser. The crystal structures of the samples were performed on an Ultima IV (XRD, Rigaku, Tokyo, Japan) with Cu Kα irradiation (λ = 0.15418 nm), and the range of diffraction angles was controlled at 5–80°. The morphology and element mapping images were taken on a Regulus 8100 (SEM, JOEL, Tokyo, Japan). The local images were taken on a 2100F (TEM, HRTEM, JOEL, Tokyo, Japan). The XPS spectra were conducted on Escalab Xi^+^ (Thermo) with Al-Kα (hν = 1486.6 eV), which were calibrated based on C1s of adventitious carbon at 284.8 eV. The diffuse reflectance UV-Vis spectra were obtained on a UV-2700 (Shimadzu, Tokyo, Japan) using BaSO_4_ as background. The nitrogen adsorption and desorption isotherm and pore size distribution were measured and analyzed by Autosorb-iQ-MP-C (Quantachrome, Norcross, GA, USA). The PL spectra were conducted on a FLS 980 (Ediburgh, livingston, Scotland, UK) using a Xenon lamp with an excitation wavelength of 375 nm at room temperature (293 K). The in-situ ESR signals were collected on an EMX plus (Bruker, Berlin, Germany) under visible-light irradiation (λ ≥ 420 nm). The photoelectrochemical measurements were carried out on PARSTAT 4000A (Princeton, NJ, USA). The ICP–MS test was measured on an Agilent 730ES (Palo Alto, CA, USA). 

### 2.4. Degradation Experiments of Tetracycline Hydrochloride

The photocatalytic activity of the catalyst was evaluated by the degradation of Tetracycline hydrochloride (TC) under visible-light irradiation. Specifically, 30 mg of photocatalyst was dispersed into a 20 mg/L 50 mL TC solution and the suspension was stirred at a certain speed. Firstly, a 4 mL reaction mixture was taken every 30 min under dark conditions to ensure that the degradation reached the adsorption–desorption equilibrium. The light source used in this experiment was a 300 W Xe lamp with a 420 nm cutoff filter. Following this step, 4 mL samples were taken every 30 min after turning on the light. The obtained reaction mixture was centrifuged, and then the concentration of the tetracycline was analyzed by measuring the absorbance at 356 nm on a UV–Vis spectrophotometer. The photo-Fenton process was operated similarly to the above photocatalytic degradation of TC, except that a concentration of H_2_O_2_ was 0.28 mol/L in the reaction system. The experiments under darkness are the same as the above steps without light.

## 3. Results and Discussion

### 3.1. The Structure of the Synthesized Samples

X-ray diffraction (XRD) patterns were carried out to determine the crystal structure of the samples. Figure 1a clearly shows five characteristic peaks located at 24.2°, 29.2°, 31.8°, 42.9° and 45.6° (JCPDS No. 037-0699), which can be indexed to (112), (11-3), (020), (105) and (422) lattice planes of pure Bi_4_O_5_Br_2_, respectively. The Fe-doped samples showed peaks at similar locations without new diffraction peaks observed. The XRD patterns demonstrated that pure Bi_4_O_5_Br_2_ and Fe-doped Bi_4_O_5_Br_2_ structures were formed. It should be noted that the peak corresponding to the (11-3) lattice plane shifts gradually to a higher angle for Fe-doped Bi_4_O_5_Br_2_ samples. This observation can be explained by the fact that a smaller radius of the Fe atom replaces the Bi atom, resulting in the distortion of the local structure of Bi_4_O_5_Br_2_. It also indicated that the Fe atoms have been doped into the Bi_4_O_5_Br_2_ framework in the form of substituting Bi atoms on the basis of Goldschmidt’s rule [37].

The molecular structure variation and the lattice distortion induced by the Fe atom doping can also be confirmed by the Raman spectra. As illustrated in Figure 1b, the characteristic peaks at 96.6 cm^−1^ and 156.3 cm^−1^ could be assigned to A_1g_ and E_1g_ of the internal Bi−Br stretching mode of Bi_4_O_5_Br_2_, respectively [38]. The peak intensity represents the content of the vibratory group, and the half-peak width is related to the number of layers in the same space [39]. It is clear that variations in the half peak width and shift in the vibrational frequency of the peak can be observed with the introduction of Fe atoms into the framework of Bi_4_O_5_Br_2_. Moreover, the intensity of peaks assigned to A_1g_ and E_1g_ modes gradually reduced with the increasing amount of Fe dopant, and even these peaks almost disappeared in the 7%Fe-Bi_4_O_5_Br_2_ sample. This result indicates that the number of Bi-Br groups decreased due to the substitution of the Fe atom doping [40]. In contrast, the widening in the half-peak of Fe-doped Bi_4_O_5_Br_2_ indicates that the Bi-Br layers in the same space reduced in number and thus the lattice distortion is formed, which resulted from the doping of Fe atoms with a smaller radius.

The specific surface area (S_BET_) and pore distribution of the samples were investigated by low temperature N_2_ physical adsorption. As shown in Figure 1c, it indicated that all samples showed a type IV N_2_ adsorption-desorption isotherm with an H_3_ hysteresis loop. This result shows that there might be a large number of active vacancies on the surface of the sample, which further confirmed that the doping of surface Fe atoms resulted in the increase in vacancies. The S_BET_ of pure Bi_4_O_5_Br_2_ is 24.6 m^2^/g, while all the Fe-doped Bi_4_O_5_Br_2_ increased in value. Among them, the S_BET_ of 7%Fe-Bi_4_O_5_Br_2_ is the largest one attaining to 75.0 m^2^/g. Similarly, the 5%Fe-Bi_4_O_5_Br_2_ increases 2.63-fold in the S_BET_ than that of pure Bi_4_O_5_Br_2_, which is 64.7 m^2^/g. The increases in the S_BET_ due to the doping of Fe atoms could offer a large number of reactive sites to participate in photodegradation reactions.

The pore size distribution of samples is mainly in the range of 2–50 nm (mesoporous) as shown in Figure 1d. Notably, the sample of 5%Fe-Bi_4_O_5_Br_2_ shows many macropores in the range of 80–140 nm. The formation of these mesopores is beneficial for the adsorption of pollutants in wastewater, and then effectively decomposed the pollutants [41]. As displayed in Table 1, it was found that the pore volume of the sample increased due to iron doping. Comparatively, the pore volume of 5%Fe-Bi_4_O_5_Br_2_ is 0.431 m^2^/g compared to that of pure Bi_4_O_5_Br_2_ (0.328 m^2^/g).

The FESEM, TEM and HRTEM are conducted to show the morphologies and the microstructure of the samples. It is obvious that both Bi_4_O_5_Br_2_ (Figure 2a and Figure 3a) and 5%Fe-Bi_4_O_5_Br_2_ (Figure 2b and Figure 3b) show nanosheet-interlaced morphologies; the thickness of 5%Fe-Bi_4_O_5_Br_2_ is close to 8.3 nm, which is much thinner than that of Bi_4_O_5_Br_2_ (about 12.5 nm). This observation indicates that the doping of Fe atoms reduced the thickness of the nanosheet, which thus reduced the transfer distance of photogenerated carriers to the surface and therefore facilitates more carriers to participate in the reaction. The lattice fringes of the samples can be clearly observed in the HRTEM of Figure 3c,d, and the lattice spacing of both Bi_4_O_5_Br_2_ and 5%Fe-Bi_4_O_5_Br_2_ are 0.28 nm, which is consistent with the index of the (11-3) lattice plane. This result indicates that Fe-doped Bi_4_O_5_Br_2_ with high crystallinity is successfully synthesized. The elemental mappings (Figure 2c) show that the O, Fe, Bi and Br elements are uniformly dispersed in 5%Fe-Bi_4_O_5_Br_2_, which indicates that the Fe element is highly distributed in the sample.

The elemental composition and states on the surface of samples are determined by X-ray photoelectron spectroscopy (XPS). As shown in Figure 4a, the obvious signals of Bi, Br and O are collected in the survey spectrum of Bi_4_O_5_Br_2_, and on the basis, an extra signal of Fe is captured on 5%Fe-Bi_4_O_5_Br_2_. In detail, the Bi 4f_5/2_ and Bi 4f_7/2_ peaks at 164.5 eV and 159.2 eV, representing the presence of Bi^3+^, are detected in the Bi 4f refined spectra for Bi_4_O_5_Br_2_ (Figure 4b). In comparison, these peaks shift ~0.15 eV to low binding energy for 5%Fe-Bi_4_O_5_Br_2_, confirming that the density of the electron cloud around the Bi atom increased. This result can be explained by the fact that the electronegativity of the Fe element (1.83) is lower than that of the Bi element (1.9) [42,43]. Therefore, the electron from the doping of Fe is easily injected into the Bi sites. As shown in Figure 4c, the Br 3d signal in Bi_4_O_5_Br_2_ can be deconvoluted into Br 3d_5/2_ and Br 3d_3/2_ at 68.6 eV and 69.6 eV, belonging to the Br^−^, respectively. Similarly, a low binding energy shift of about 0.15 eV occurred in the Br 3d spectrum of 5%Fe-Bi_4_O_5_Br_2_. Therefore, the electron density of the Br atom plate is stronger, which is affected by the doping of Fe possessing more valence electrons compared to the Bi atom [44]. Furthermore, in the high-resolution O 1s spectra (Figure 4d), two peaks at 529.8 eV and 531.2 eV are observed in Bi_4_O_5_Br_2_ and 5%Fe-Bi_4_O_5_Br_2_, which are ascribed to the intrinsic lattice oxygen and absorbed oxygen [9]. In addition, Bi_4_O_5_Br_2_ possessed an additional peak at 533.1 eV assigned to the absorbed water [45]. Deeply, the signal of Fe 2p can be fitted into four peaks for 5%Fe-Bi_4_O_5_Br_2_ corresponding to Fe 2p_1/2_ and Fe 2p_3/2_, each of which contains two different valence states of Fe (Figure 4e). Among them, the two peaks at 725.8 eV and 712.5 eV are assigned to the Fe^3+^. Meanwhile, the other two peaks at 722.6 eV and 710.0 eV are assigned to the Fe^2+^ [45,46]. In terms of contributing to the Fe 2p, the content of Fe^2+^ accounted for 0.587%, which is larger than Fe^3+^ (0.413%), indicating the surface content of Fe^2+^ is higher. Therefore, the abundant redox Fe^2+^/Fe^3+^ on the surface are facilitated to excite H_2_O_2_ to boost the degradation performance of the TC.

The photophysical properties of the samples are measured by UV–vis diffuse reflectance spectroscopy. As shown in Figure 5a, the absorbance edge of pure Bi_4_O_5_Br_2_ is around 500 nm, indicating that this sample is a visible-light responsive narrow bandgap photocatalyst. It is obvious that the red-shift in the absorbance edge with the introduction of the Fe atom can be observed, demonstrating that the absorbance spectrum in the range of visible light is broadened with the increasing amount of dopant. In addition, the band gap of Fe-Bi_4_O_5_Br_2_ becomes narrow compared to that of pure Bi_4_O_5_Br_2_, reflected in the Tauc plot (Figure 5b). The bandgaps of Bi_4_O_5_Br_2_, 1%Fe-Bi_4_O_5_Br_2_, 3%Fe-Bi_4_O_5_Br_2_, 5%Fe-Bi_4_O_5_Br_2_ and 7%Fe-Bi_4_O_5_Br_2_ were calculated to be about 2.26, 1.87, 1.91, 1.71 and 1.49 eV, respectively. The doping of Fe atoms may result in the formation of an intermediate state energy level, thus narrowing the bandgap of the sample [47,48]. The flat-band potentials of the samples were measured by the electrochemical method reflected in the Mott-Schottky plots (Figure 5c). Both samples showed curves of *n*-type semiconductors in which the slope of the fitted linear graph is positive. The flat-band potential of pure Bi_4_O_5_Br_2_ and 5%Fe-Bi_4_O_5_Br_2_ are −0.42 eV and −0.20 eV (vs. Ag/AgCl), respectively. The conduction band (CB) position is at a negative 0.1~0.2 eV compared to the flat-band potential for N-type semiconductors [49]. Thus by Equation (1), the CB of pure Bi_4_O_5_Br_2_ and 5%Fe-Bi_4_O_5_Br_2_ are −0.43 and −0.21 eV, respectively (vs. NHE). The valance band (VB) position is calculated via Equation (2), and finally, the distinct band structure is plotted in Figure 5d. In detail, the 5%Fe-Bi_4_O_5_Br_2_ gives a more positive conduction band compared to the single-electron reduction in O_2_ (E [O_2_/^•^O_2_^−^] = −0.33 eV vs. NHE). Moreover, based on the bandgap of 1.71 eV, the VB potential of 5%Fe-Bi_4_O_5_Br_2_ is 1.50 eV (vs. NHE), which is more negative than that of Bi_4_O_5_Br_2_ (1.83 eV vs. NHE).
(1)$$E_{\mathrm{NHE}}=E_{\mathrm{Ag}/\mathrm{AgCl}}+0.0591\times \mathrm{pH}+E_{\mathrm{Ag}/\mathrm{AgCl}\;}^{0};E_{\mathrm{Ag}/\mathrm{AgCl}}^{0}=0.1976V\;\mathrm{vs}.\;\mathrm{NHE}\left(25\;{}^{\circ}C\right)$$
(2)$$E_{VB}=E_{g}+E_{CB}$$

Figure 6a shows the transient photocurrent density response of Bi_4_O_5_Br_2_ and Fe-doped Bi_4_O_5_Br_2_ samples. It can be observed that the series of Fe-doped samples produce intensive current responses compared to Bi_4_O_5_Br_2_ under visible-light irradiation. The photocurrent intensity of the 5%Fe-Bi_4_O_5_Br_2_ photocatalyst exhibited much stronger than that of pure Bi_4_O_5_Br_2_. The enhanced photocurrent confirmed that the effective separation of the photogenerated electron−hole pairs is driven by a boosted internal electric field generated by the doping of Fe. Furthermore, the transfer resistance of carriers within the samples could be evaluated from the electrochemical impedance spectroscopy (EIS). Generally, the shorter semi-cycle arc radius in the Nyquist plot means a lower resistance to the charge transfer. Therefore, as shown in Figure 6b, the semicircular radius in the Nyquist plots of the Fe-Bi_4_O_5_Br_2_ series samples decreases with the introduction of Fe, which confirmed that the transfer resistances of carriers in the Fe-Bi_4_O_5_Br_2_ series samples are less than that in Bi_4_O_5_Br_2_. The semi−cycle arc radius in the Nyquist plot of 5%Fe-Bi_4_O_5_Br_2_ is the shortest one, indicative of the quickest carrier transferring rate of this sample being generated. In summary, the doping of Fe atoms into the framework of Bi_4_O_5_Br_2_ resulted in the occurrence of spontaneous polarization and therefore enhancement in the internal electric field, which effectively induced the separation and rapid migration of the photogenerated electron–hole pairs [50].

The recombination kinetics of photogenerated carriers can be comprehensively simulated by the measurement of fluorescence spectra. In the steady-state photoluminescence (PL) spectra (Figure 6c), 5%Fe-Bi_4_O_5_Br_2_ responds with a weaker fluorescence intensity compared to Bi_4_O_5_Br_2_ upon excitation at a 375 nm exciting light source. The result indicated that the recombination of photogenerated carriers was inhibited in 5%Fe-Bi_4_O_5_Br_2_, which is consistent with the result observed in the photocurrent. In addition, the average fluorescence lifetime (τ) of photogenerated carriers is evaluated in the time−resolved photoluminescence spectra (TRPL). As shown in Figure 6d, the τ value of 5%Fe-Bi_4_O_5_Br_2_ is 3.112 ns, which is effectively extended than that of Bi_4_O_5_Br_2_ (2.909 ns), suggesting that facilitated carriers with longer residence time are generated in the sample. These results further demonstrated that the photogenerated carriers can be effectively separated and transferred driven by the built−in electron field enhanced by 5%Fe-Bi_4_O_5_Br_2_. 

### 3.2. Degradation Performance of TC

The photocatalytic activities of the Bi_4_O_5_Br_2_ and Fe-Bi_4_O_5_Br_2_ samples were investigated by the degradation of TC under visible-light irradiation. As shown in Figure 7a, it is clear that almost no loss of TC in the absence of the photocatalyst in the control experiment can be observed, which confirmed that the degradation of TC is entirely derived from the action of the photocatalysts. Moreover, a dark treatment for 1 h is enough for the TC to reach an adsorption–desorption equilibrium on the surface of all the samples. The 5%Fe-Bi_4_O_5_Br_2_ sample shows the strongest adsorption for TC, almost three times as much as Bi_4_O_5_Br_2_, which is attributed to a larger specific surface area exposing more adsorption sites. Significantly, 5%Fe-Bi_4_O_5_Br_2_ possesses the best photocatalytic degradation activity for TC under visible-light irradiation for 3 h, reaching 60%. However, a further increase in the amount of the dopant into Bi_4_O_5_Br_2_ induces an adverse effect on the photocatalytic performance. It can be observed that the photocatalytic activity of 7%Fe-Bi_4_O_5_Br_2_ for the removal of TC decreases to only 28.9%, even worse than that of pure Bi_4_O_5_Br_2_ (46.6%). Therefore, the appropriate ratio of Fe doping can induce the spontaneous polarization and enhance the built-in electric field, thus achieving effective separation of the photogenerated carrier. Moreover, the photocatalytic degradation of TC is evaluated in the first-order reaction by Equation (3). Further, as shown in Figure 7b, the reaction rate constant (k) of 5%Fe-Bi_4_O_5_Br_2_ is fitted to 0.006 min^−1^, which increases nearly 1.5-fold than that of Bi_4_O_5_Br_2_ (0.004 min^−1^).
(3)$$\mathrm{In}\;C_{t}=\mathrm{In}C_{0}-k_{1}t$$

Furthermore, the photo−Fenton system is employed to improve the degradation efficiency of TC, and the dependence of the activity on the structure of the photocatalyst in the presence of H_2_O_2_ is studied. As shown in Figure 7c, without the photocatalyst while only the H_2_O_2_ system is added, 29.4% of TC degradation activity is achieved under visible-light irradiation. It is interesting that when the Fe-Bi_4_O_5_Br_2_ series of samples are supplied, the degradation performance of TC in the presence of H_2_O_2_ under visible-light irradiation is promoted significantly and is much higher than that of Bi_4_O_5_Br_2_ with the H_2_O_2_ system (57.9%). The highest photocatalytic activity over the optimized sample, 5%Fe-Bi_4_O_5_Br_2_, can reach the entire removal of TC in 3 h under visible-light irradiation. Similarly, the degradation reaction rate constant of TC in the photo−Fenton system is also estimated according to the first-order reaction. As shown in Figure 7d, the highest k is 0.016 min^−1^ for the 5%Fe-Bi_4_O_5_Br_2_-Fenton system, which is almost 3.2 times greater compared to the Bi_4_O_5_Br_2_-H_2_O_2_ system (0.005 min^−1^). This result indicates that the optimized recombination kinetics of the photogenerated carrier over 5%Fe-Bi_4_O_5_Br_2_ facilitates to activate H_2_O_2_ sufficiently to generate strong oxidation ^•^OH and expedites the degradation reaction kinetic.

As shown in Figure 7e, the Fenton process for the degradation of TC is evaluated under darkness. Specifically, almost 17% of TC is removed only with H_2_O_2_, and the Bi_4_O_5_Br_2_-H_2_O_2_ shows the degradation ratio of TC is about 27%, which is comparable to that of only with H_2_O_2_ regardless of the adsorption capacity of the photocatalyst. Apparently, the degradation performance is enhanced in the 5%Fe-Bi_4_O_5_Br_2_-H_2_O_2_ Fenton system (42%). It demonstrates that the 5%Fe-Bi_4_O_5_Br_2_ could activate H_2_O_2_ via the exposed Fe ions under darkness, whereas Bi_4_O_5_Br_2_ could not.

From the above results, the 5%Fe-Bi_4_O_5_Br_2_ photocatalyst showed excellent activity with light and H_2_O_2_. Finally, in order to study the photocatalytic stability of the catalyst, a cyclic experiment was carried out, as shown in Figure 7f. After five cycles for the degradation of TC, the 5%Fe-Bi_4_O_5_Br_2_ still retains its excellent performance (85%), with only a 15% loss of degradation ratio compared to the fresh reaction. In addition, it is found that without obvious variations of the crystal structure of the samples after five cycles can be observed. Moreover, the concentration of Fe ions in the solution was negligible (about 0.022 mg/L) in the used 5%Fe-Bi_4_O_5_Br_2_ after the cycle experiments in Figure 7h.

In order to explore the active species over Fe-Bi_4_O_5_Br_2_ in the photocatalytic process, an in-situ ESR technique is employed. The radical capture reagent TEMPO (2,2,6,6-tetramethyl-1-piperidinyloxy) is utilized in the photocatalytic system to remove the h^+^. As shown in Figure 8a,b, the triple peak in the ESR signal shows the characteristic species of TEMPO with h^+^. The decreases in the peak intensity represent that the generated h^+^ is captured by TEMPO. It is obvious that the decrease in peak intensity of the 5%Fe-Bi_4_O_5_Br_2_ is more prominent than that of the Bi_4_O_5_Br_2_ after 10 min visible-light irradiation, indicating that the 5%Fe-Bi_4_O_5_Br_2_ may be able to generate holes more efficiently during the photocatalytic degradation. The generated ^•^OH is captured by the 5,5-dimethyl-1-pyrroline (DMPO) in the 0.28 M H_2_O_2_ aqueous solution. As shown in Figure 8c,d, four characteristic peaks with an intensity of 1:2:2:1 are collected, demonstrating that the Bi_4_O_5_Br_2_ and the 5%Fe-Bi_4_O_5_Br_2_ can activate H_2_O_2_ efficiently under visible-light irradiation [51]. Additionally, the more intensive signal of DMPO confirmed that more photogenerated carriers in the 5%Fe-Bi_4_O_5_Br_2_ could be used to activate H_2_O_2_ to produce a greater number of ^•^OH.

As shown in Figure 8e, the characteristic signal of DMPO-^•^O_2_^−^ was not observed on the 5%Fe-Bi_4_O_5_Br_2_ regardless of the irradiation or not. This result indicated that the ^•^O_2_^−^ species could not be generated through the single−electron reduction in oxygen under visible-light irradiation; for that the CB potential of Fe-Bi_4_O_5_Br_2_ is −0.21 eV vs. NHE (E [O_2_/^•^O_2_^−^] = −0.33 eV vs. NHE). Interestingly, the ESR signal of ^•^O_2_^−^ can be observed with 10 min irradiation in the presence of H_2_O_2_ in Figure 8f. It is therefore demonstrated that the ^•^O_2_^−^ species could be generated by the activation of H_2_O_2_ through excited 5%Fe-Bi_4_O_5_Br_2_. So, both the ^•^OH and ^•^O_2_^−^ could be generated by visible-light irradiation of the Fe-Bi_4_O_5_Br_2_ photocatalyst to realize the efficient degradation of tetracycline.

In summary, combined with the above discussion of free radicals, the mechanism and diagram can be presumed and shown in Equations (4)–(10) and Figure 9. Under the irradiation of visible light, the carriers are separated and shifted efficiently to the photocatalyst surface (Equation (4)). Due to the enhancement in the spontaneous polarization effect, a great number of photogenerated charges reach the surface of 5%Fe-Bi_4_O_5_Br_2_. More holes are transferred to the surface of the 5%Fe-Bi_4_O_5_Br_2_ photocatalyst from the in−situ ESR spectrum. Photogenerated holes may oxidize TC (Equation (5)), while the Fe^3+^ species is reduced to Fe^2+^ by the photogenerated electrons (Equation (6)). Moreover, more Fe(II) sites were synthetized on the 5%Fe-Bi_4_O_5_Br_2_ from the XPS spectrum_._ Due to the addition of hydrogen peroxide, the Fenton effect in the reaction, that is, Fe^2+^ reacts with H_2_O_2_ to form Fe^3+^ and produce ^•^OH (Equation (7)), and generated ^•^OH can degrade TC (Equation (9)). Finally, Fe^3+^ combined with hydrogen peroxide will produce ^•^O_2_^−^ (Equation (8)) [52], which can further degrade TC pollutants (Equation (10)).
(4)$$\mathrm{Photocatalyst}+\mathrm{hv}\to h^{+}+e^{-}$$
(5)$$h^{+}+\mathrm{TC}\to \mathrm{Degradation}\;\mathrm{products}\to H_{2}O+{\mathrm{CO}}_{2}$$
(6)$$e^{-}+{\mathrm{Fe}}^{3+}\to {\mathrm{Fe}}^{2+}$$
(7)$${\mathrm{Fe}}^{2+}+H_{2}O_{2}\to {\mathrm{Fe}}^{3+}+∙\mathrm{OH}+{\mathrm{OH}}^{-}$$
(8)$${\mathrm{Fe}}^{3+}+H_{2}O_{2}\to {\mathrm{Fe}}^{2+}+∙O_{2}^{-}+2H^{+}$$
(9)$$∙\mathrm{OH}++\mathrm{TC}\to \mathrm{Degradation}\;\mathrm{products}\to H_{2}O+{\mathrm{CO}}_{2}$$
(10)$$∙O_{2}^{-}+\mathrm{TC}\to \mathrm{Degradation}\;\mathrm{products}\to H_{2}O+{\mathrm{CO}}_{2}$$

## 4. Conclusions

In conclusion, the photo-Fenton degradation system was constructed via Fe-doped Bi_4_O_5_Br_2_ nanosheet. The obtained 5%Fe-Bi_4_O_5_Br_2_-photo-Fenton system realized excellent degradation performance (almost 100%) for the removal of Tetracycline under visible-light irradiation. The apparent reaction rate constant of 5%Fe-Bi_4_O_5_Br_2_ reaches 0.016 min^−1^, which is almost 3.2 times faster than that of Bi_4_O_5_Br_2_ (0.005 min^−1^). The complete removal of Tetracycline and the enhanced reaction rate are primarily attributed to the multiple effects of Fe doping in the 5%Fe-Bi_4_O_5_Br_2_-photo-Fenton system. On one hand, the doping of Fe induces the spontaneous polarization, which enhances the built-in electric field to modulate the photogenerated carrier separation kinetics of 5%Fe-Bi_4_O_5_Br_2_. On the other hand, the Fe(II) in the molecular framework becomes the activation center of H_2_O_2_ and the abundant photogenerated carriers accelerate the transition between Fe(II) and Fe(III), thus efficiently activating H_2_O_2_ to generate ^•^O_2_^−^ and more ^•^OH. The holes also play a decisive role to oxidize Tetracycline under visible-light irradiation. These active species are confirmed by in-situ ESR. Therefore, the 5%Fe-Bi_4_O_5_Br_2_-photo-Fenton system is a potential candidate for “green” photocatalytic removal of antibiotics.

## Data Availability

No new data were created or analyzed in this study. Data sharing is not applicable to this article.

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
