# Peer review of "Enhanced Photo-Assisted Fenton Degradation of Antibiotics over Iron-Doped Bi-Rich Bismuth Oxybromide Photocatalyst"

_nanomaterials, 2022, doi:10.3390/nano13010188_

Round 1

Reviewer 1 Report

The photoFenton process is not new but widely studied. The literature references are few and too much recent. In the introduction part this should be changed.

The calcution of kinetics constants for photocatalytic degradation should be reported and showed.

The authors have then applied a photoFenton process to TC degradation. They simply verified with the addition of an unknown amount of H2O2 the further degradation achieved. The process should be further clarified in its mechanism.

In particular it's no clear if the iron goes in solution and gives rise to an homogenous Fenton process. So the authors should estabilish if the perocess is heterogeneous or homogenous. Furthermore the leaching of iron from the photocatalyst should be verified. The TC degradation tests are ambiguos. 

Despite a large and quite well selected characterization performed , some affirmations are quite surprisingly like "the thickness of 5%Fe-Bi4O5Br2 is closed to 8.3 nm, which is much thinner than that of 189 Bi4O5Br2 (about 12.5 nm)" Where is this estimation?

In addition, their  partial conclusion "The elemental mappings (Figure. 2c) shows 196 that the O, Fe, Bi and Br elements are uniformly dispersed in 5%Fe-Bi4O5Br2, which indi- 197 cated that the Fe atoms are highly dispersed on the surface of the sample." leads to consider that iron species are formed on the surface. These species must be identified.

For these reasons a deep revision of the manuscript should be performed, being not publishable in the present form.

Author Response

Thank you for your suggestions. Please find the reply of your questions in the attached file.

Reviewer 2 Report

The manuscript submitted for consideration in Nanomaterials entitled “Enhanced photo-assisted Fenton degradation of antibiotics over 2 iron-doped Bi-rich bismuth oxybromide photocatalyst” by Fengjiao Zhang, Yanhua Peng, Xiaolong Yang, Zhuo Li and Yan Zhang is from the beginning a complex work, and actually the title awakes the reader about this. It is difficult to be completely sure of what is doing each agent, including Fe, Bi, with the oxybromide moieties, and then the role of the photocatalysis for the degradation of antibiotics, apart from the use of H2O2 that is another fact of uncertainty. This is the main concern, the authors should do efforts to clarify why they are sure of the role of each agent, by means of the blank test that could be performed, otherwise it would be necessary to express that they are not convinced, and it is a simple hypothesis. Actually, at least with and without Fe for example. And also of the light.

The characterization is detailed and is consistent.

The references should be amended because the format of the journal titles is difference. Take for instance for reference 23, or reference 34.

Minor points:

Replace “Fe-doped Bi4O5Br2 nanosheets was synthesized  by “Fe-doped Bi4O5Br2 nanosheets were synthesized

Improve strange expressions such as “Raman spectrum was conducted

Correct all times that Figures are mentioned throughout the text, take for instance “Figure. 2a and Figure. 3a”, where the dot after “Figure” should be removed.

Replace “it 294 is clearly that almost  by  it 294 is clear that almost

Replace “It is obviously that the decrease  by  It is obvious that the decrease

Overall, if the authors are able to improve the format, but specially to clarify the role of each agent, the paper could be considered for publication in Nanomaterials.

Author Response

Thank you for your good questions! We have answered them in the attached file.

Round 2

Reviewer 1 Report

The paper appears improved, altough the reason of the selection of H2O2 concentration is not reported. However, the authors have not added into the mechanism the regeneration of Fe2+ in the catalytic cycle induced by the light. Fe3+ + H2O+ hv >  Fe2+ + HO + H+

They have to deep the photoFenton study. 

Moreover, some of their kinetics curves don't pass through zero, a mathematical condition that has to be respected in the pseudo-first order modelling. 

Author Response

The paper appears improved, although the reason of the selection of H2O2 concentration is not reported. However, the authors have not added into the mechanism the regeneration of Fe2+ in the catalytic cycle induced by the light. Fe3+ + H2O+ hv >  Fe2+ + HO  + H+, They have to deep the photoFenton study.

Reply: Thank you for your suggestion. Fe3+ can be reduced to Fe2+ under the irradiation of ultraviolet light according to the following reaction: Fe3+aq+H2O+hv → Fe2+aq+H++•OH. It is a homogeneous reaction. However, since the catalytic system studied in this manuscript is a heterogeneous catalytic system, the mechanism for the regeneration of Fe2+ in the catalytic cycle should be different with above homogeneous system. Therefore, we would like to discuss the mechanism in more detail in the next paper.

 Moreover, some of their kinetics curves don't pass through zero, a mathematical condition that has to be respected in the pseudo-first order modelling. 

Reply: Thank you for your advice. According to your instruction, the kinetics curves are analyzed in a more reasonable mode (pass through zero). Please find it in the revised manuscript.

Reviewer 2 Report

The first revision of the manuscript submitted for consideration in Nanomaterials entitled “Enhanced photo-assisted Fenton degradation of antibiotics over 2 iron-doped Bi-rich bismuth oxybromide photocatalyst” by Fengjiao Zhang, Yanhua Peng, Xiaolong Yang, Zhuo Li and Yan Zhang has addressed all my concerns. I only regret a bit not having a more detailed answer for each concern. Instead I went through the manuscript where the changes are highlighted.

In the revision of the paper by the editorial staff, the journal abbreviations will be used instead of the full names that should be in capitals (first letter) but since the format is the abbreviated, this is not an issue.

I suggest publication as it is.

Author Response

Thank your for your comments.

Round 3

Reviewer 1 Report

The kinetics constant evaluation has been improved. However, this model does not fit well the experimental data. 

The mechanism and the scheme in fig. 9 have to agree

Author Response

Response to Reviewer 1 Comments

Point 1. The kinetics constant evaluation has been improved. However, this model does not fit well the experimental data. 

Reply: Thank you for your suggestion. As it is a heterogeneous catalytic system, most of the experimental data can be modulated to fit the model, which is closer to the first-order kinetics. So, the photocatalytic degradation of TC is evaluated in the first-order reaction by Eq. 3, which is shown in the manuscript. We have deleted the sentence “Unambiguously the photocatalytic degradation of TC over samples is in line with a first-order reaction kinetic.” (line 314~)

Point 2. The mechanism and the scheme in fig. 9 have to agree

Reply: Thank you for your advice. In fact, the mechanism is agreed to the scheme in fig. 9. In order to express it more clearly, we added the number of the corresponding equation in the revised manuscript. Moreover, the corresponding expressions can also be found in fig. 9.
